# An Assessment of the Freshness of Fruits and Vegetables Through the Utilization of Bioimpedance Spectroscopy (BIS)—A Preliminary Study

**DOI:** 10.3390/foods14060947

**Published:** 2025-03-11

**Authors:** Mirella Kluza, Ilona Karpiel, Klaudia Duch, Dariusz Komorowski, Szymon Sieciński

**Affiliations:** 1Łukasiewicz Research Network—Krakow Institute of Technology, ul. Zakopiańska 73, 30-418 Krakow, Poland; mirella.kluza@kit.lukasiewicz.gov.pl; 2Faculty of Biomedical Engineering, Department of Medical Informatics and Artificial Intelligence, Silesian University of Technology, Roosevelt 40, 41-800 Zabrze, Poland; dariusz.komorowski@polsl.pl; 3Faculty of Science and Technology, Institute of Biomedical Engineering, The University of Silesia in Katowice, 75 Pułku Piechoty 1a, 41-500 Chorzów, Poland; klaudia.duch@smcebi.edu.pl; 4Academy of Silesia, Department of Clinical Engineering, Rolna 43, 40-555 Katowice, Poland; szymon.siecinski@akademiaslaska.pl; 5Institute of Medical Informatics, University of Lübeck, Ratzeburger Allee 160, 23562 Lübeck, Germany

**Keywords:** bioimpedance, fruits, vegetables, ripeness, freshness

## Abstract

This study evaluates the use of bioimpedance spectroscopy (BIS) as a non-invasive method to assess the freshness of fruits and vegetables by measuring impedance, its components, and phase angle. Over nine days, three vegetables (potato, pumpkin, and red pepper) and two fruits (apple and banana) were assessed using the Analog Discovery 3 device, covering a frequency range of 50 Hz to 1 MHz. The results showed a consistent decrease in impedance and an increase in phase angle during ripening, with statistical significance observed for pumpkin and potato (*p* < 0.05). The findings confirm BIS as an effective, objective, and non-destructive alternative to traditional chemical methods for monitoring freshness, despite challenges such as structural damage in red pepper. This integration of BIS into food quality assessment and healthcare provides a multidisciplinary approach to improving nutrition and health.

## 1. Introduction

Recently, increasing emphasis has been placed on maintaining a healthy lifestyle and following a balanced diet. Consumers’ awareness and expectations regarding the quality of the products they consume are increasing, especially in the case of fruits and vegetables. One of the key factors of a product is its freshness, which affects its taste and nutritional value, as well as its shelf life and health safety. Traditional methods used to assess the freshness of fruits and vegetables are mainly based on subjective visual and sensory evaluations. Among the various techniques for assessing the quality of fruits and vegetables are non-invasive methods. The most frequently used ones include hyperspectral imaging, Raman imaging, MRI imaging, laser backscattering imaging, and thermal imaging [1]. They enable the assessment of the internal structure and the prediction of the chemical composition of the studied object. This translates into a more accurate evaluation of freshness without incurring losses. Non-invasive imaging techniques for assessing freshness are also helpful because of the measurements’ simplicity and the results’ reliability. However, these methods do have drawbacks. The tests take a long time, which increases production and distribution costs. The main challenge is still the practical application of the available techniques. Time-consuming data acquisition, high costs, high interference from the materials of the object and the environment, and the lack of unique spectral characteristics for different fruits and vegetables are issues that need to be looked at in the context of developing non-invasive methods for assessing the freshness of the produce [2].

A promising technique that can be used to analyze the freshness of fruits and vegetables is bioimpedance spectroscopy (BIS). This is a technique that allows the measurement of the electrical impedance of biological samples over a wide range of frequencies. As BIS deploys impedance change depending on the physiochemical characteristics of the sample, the method can be used to monitor health [3,4], body composition [5,6] and the quality of food produce [2,7]. BIS enables the analysis of physicochemical changes, such as a decrease in impedance and an increase in phase angle in the ripening process. The high statistical agreement of results confirms the effectiveness of this technique, although in some cases, such as with red peppers, disruptions related to intense decomposition may occur. Vegetables and fruits can be sources of electrical properties such as electrical conductivity and electrical permittivity. Electrical conductivity describes how well the substance transmits electric current in its structure and to what extent it is affected by the chemical composition of materials and tissue structure [8]. Electrical permittivity in fruits, vegetables, and other agricultural produce is related to cell integrity, chemical composition, and moisture content [9]. Hence, the correlation between the above-described electrical properties and factors such as temperature, sugar content, firmness, and storage time of the food produced under study can be determined and evaluated. The interest in the electrical conductivity of food is gradually increasing along with the advancement in food processing technology, food quality evaluation, and food preservation [10].

This article focuses on the potential use of bioimpedance spectroscopy to assess the freshness of fruits and vegetables by analyzing their physiochemical parameters. The state-of-the-art BIS applications in food quality analysis are presented, and the advantages and limitations of the technique and prospects for further development in improving fruit and vegetable freshness assessment are discussed. The study’s main objective was to evaluate and establish relationships between the measured parameters. Due to the different sizes and geometries of the selected fruits and vegetables, the parameters can be analyzed depending on the placement of the electrodes.

Bioimpedance spectroscopy (BIS) has recently gained significant attention as a non-destructive food quality analysis technique. This method measures the electrical impedance of food products over a range of frequencies, providing valuable information about their composition, structure, and quality. According to recent studies, BIS has been effectively used to assess the freshness and ripeness of fruits and vegetables by monitoring changes in their electrical properties. For instance, BIS has been employed to evaluate the ripening stages of tomatoes [11], apples [12], and bananas [13], demonstrating a strong correlation between impedance measurements and biochemical changes during ripening. Moreover, BIS has shown promise in the analysis of meat products. It has been used to determine the water and fat content [14,15] and overall meat quality [16,17], providing a rapid and accurate assessment without the need for destructive sampling. Studies have also highlighted the potential of BIS in detecting spoilage and microbial contamination in meat [18], ensuring food safety and extending shelf life. BIS has also been utilized in the dairy industry to analyze the composition of milk and dairy products. It has been used to measure the fat and protein content [19] and overall milk and dairy product quality [20,21], offering a non-invasive and efficient alternative to traditional methods. Additionally, BIS has been applied to detect adulteration in dairy products [22,23], ensuring product authenticity and consumer safety.

## 2. Materials and Methods

### 2.1. Preparation of Materials

Three vegetables were selected for testing (pumpkin, red pepper, and potato), along with two fruits (banana and apple). The materials were selected to encompass a range of sizes, shapes, and water-to-flesh ratios. Special attention was also paid to the selection of products with different thicknesses and skin firmness due to the specificity of performing bioimpedance measurements. Selected fruits and vegetables were purchased from a local vegetable store and washed with water. Throughout the study, they were stored in a cool room at a temperature of 16 °C and a humidity of 35%. Constant storage conditions throughout the experiment ensure that external variables affecting the results are minimized.

### 2.2. Measurement Equipment and Configuration

Impedance studies of fruits and vegetables were carried out using the Analog Discovery 3 device and the dedicated software provided by Digilent (Pullman, WA, USA)—WaveForms v3.20.1. Analog Discovery 3 is an advanced signal measurement and analysis device that combines the functions of several different tools—digital oscillator, waveform generator, logic analyzer, pattern generator, and variable power supply. For BIS measurements, the Impedance Analyzer was used, which allows for the integration of the signal generator and oscilloscope functions, in addition to an external reference resistor. The hardware setup scheme is presented in Figure 1. The Impedance Analyzer can measure impedance, admittance, inductance, and capacitance and then present these values in different charts as Nyquist or Bode plots. The Analog Discovery 3 specifications report that the device offers an accuracy of ±0.5% for impedance measurements.

### 2.3. Bioimpedance Measurement

Tests on the selected fruits and vegetables were conducted from 20 November 2023 to 1 December 2023, with one test each day (ten tests for each) (Figure 2).

Two electrodes were glued to the selected fruits and vegetables on both sides of each fruit and vegetable. Standard ECG electrodes and standard ECG leads were used for the measurements. The leads were rearranged so that they could be connected to the Analog Discovery 3 (see Figure 3). One of the leads was connected to the output of the signal generator and channel one of the Analog Discovery 3 oscilloscope, and the other lead was connected to channel two of the oscilloscope and to the ground through a resistor load. Resistance values of the resistor were chosen based on measurements with an ohmmeter of all fruits and vegetables. First, the ohmmeter was applied to the fruit/vegetable, and the resistance was measured; then, the resistor with the closest resistance value shown on the ohmmeter was selected. For the potato, pumpkin, banana, and apple, a 10 kΩ resistor was selected, and for the red pepper, a 1 MΩ resistor was used. The measured parameters were impedance, phase angle, and the real part and imaginary part of the impedance. The waveforms of the signals were also registered. The frequency range of the conducted tests was 50 Hz–1 MHz for 201 measurement points. The values were selected based on the expert scientific literature [7,24,25].

### 2.4. Statistical Analysis

Statistical analysis was performed with Statistica 13.1 software. After checking the normal distribution (tested with the Shapiro–Wilk test) and the homogeneity of variance (Leven’s test), the data did not meet the assumptions required for the parametric test; the non-parametric Friedman ANOVA test was used. The level of statistical significance was set at *p* < 0.05.

## 3. Results

The results of the conducted studies reveal the physicochemical phenomena occurring in the tested fruits and vegetables during 9 days of storage, using bioimpedance spectroscopy. During the measurement period, the red bell pepper exhibited the most significant decomposition, rupturing on the fourth day. Consequently, it was excluded from the study due to the distortion of measurement results. The other test subjects, including the banana, potato, pumpkin, and apple, showed varying degrees of decomposition. However, none of these exhibited substantial damage that could potentially influence the overall course and outcomes of the experiment.

Table 1 shows the impedance values of banana, pumpkin, apple, potato, and bell pepper for selected frequencies: 100 Hz, 1 kHz, 10 kHz, 100 kHz, and 1 MHz. Figure 4 shows the impedance changes for each fruit and vegetable on each measurement day in the entire frequency spectrum (50 Hz–1 MHz, 201 consecutive measurement points). In both the table and the graph, it can be observed that the impedance values decrease with the days of storage and the increase in signal frequency. The observed trends are particularly pronounced at low frequencies (100 Hz and 1 kHz), suggesting that these frequency ranges may be crucial for assessing the freshness of products. Statistical analysis of the results using Friedman’s ANOVA test showed significant differences in impedance values between individual days (*p* = 0.000009), indicating the high sensitivity of the BIS method in monitoring the ripening processes. The consistency coefficient for impedance (0.75582) shows a very high similarity of variables, which indicates very high similarity in terms of impedance properties, i.e., electrical properties of tested vegetables and fruits. This may be due to similar water content, cellular structure, ionic conductivity, or other physicochemical characteristics. A similar impedance may indicate a similar degree of ripeness, freshness, or water content. The result indicates that in terms of impedance, these fruits and vegetables belong to the same group (high water content and typical cell structure).

### 3.1. Changes in Phase Angle

The phase angle, which indicates changes in the distribution of intra- and extracellular water, systematically increased during the successive days of measurements for all tested products except for the bell pepper, whose results were distorted due to skin cracking (Figure 5).

In the case of bananas, the phase angle values increased from −36.12° to −29.09°; for pumpkin, from −40.71° to −29.95°; for potato, from −53.00° to −34.65°; and for apple, from −49.84° to −43.21°. This increase reflects progressive changes in cell membrane permeability and the degradation of cellular structures, confirming the effectiveness of BIS as a monitoring tool for cellular-level transformations during the ripening process. Figure 6 illustrates the relationship between phase angle and impedance for the tested samples. The greatest differences are observed for red pepper, which could be attributed to the structural cracking of the vegetable caused by water loss and internal pressure changes. For the remaining fruits and vegetables, a consistent trend of increasing phase angle with decreasing impedance is evident, which is likely linked to the natural ripening process and the progressive breakdown of cell membranes over time.

The Bode plots for all the tested fruits and vegetables (Figure 7) reveal a characteristic phase shift across the frequency spectrum. The observed phase angles, ranging from −60° to −20°, correspond to a combination of resistive and capacitive behaviors, which are typical for biological tissues undergoing degradation. Notably, the steepest changes occur at low frequencies, reinforcing the conclusion that this frequency range is the most sensitive to ripening and freshness loss changes. Figure 8 presents the Nyquist plots for the analyzed samples, highlighting the relationship between the real and imaginary parts of the impedance. The observed semicircular patterns reflect cell membranes’ capacitive properties, while the plots’ shifts over time suggest progressive damage to cellular structures. The shrinking of the semicircles indicates a decrease in membrane integrity, which is especially noticeable for the banana and red pepper. Overall, analyzing phase angle changes across different frequencies and the corresponding impedance behavior provides a comprehensive insight into the ripening dynamics of fruits and vegetables.

The results highlight that BIS is particularly effective in identifying subtle physiological changes, such as those occurring in thick-skinned vegetables like pumpkin and potato, where external signs of degradation are not immediately visible. The results suggest progressive changes in cell membrane permeability and confirm that BIS is an effective tool for monitoring structural changes at the cellular level. A novelty in this study was the detailed analysis of the phase angle for thick-skinned vegetables such as pumpkin and potato. It was shown that the increase in phase angle correlates with the degradation of the internal tissue structure despite the lack of visible external changes. The phase angle measures the tendency of dielectric materials to absorb and dissipate energy without inducing charge alignment on the membrane surface. This reflects the distribution of intra- and extracellular water, and reduced values may be related to changes in membrane permeability. Fresh fruits and vegetables are usually characterized by firm and taut skin, intense color, and a fresh smell. Warning signs include loss of firmness, wrinkles, a malodorous fermentation smell, brown spots on the surface, and mold on the skin, indicating that the freshness of the product has expired.

### 3.2. Equivalent Circuit

Equivalent circuit modeling is achieved using EIS Spectrum Analyser v1.0 software. The best fit model, the statistics of the model, the equivalent electrical circuit, and the parameters are shown in Figure 9. The equivalent circuit parameters are obtained based on the impedance curve fitting in the software. The values of the equivalent circuit parameters were obtained based on the banana tissue from the last day of measurements. The values of the parameters will differ in connection with the fruit/vegetable, the different storage conditions, the variety, etc. The components employed in circuit design adhere to the existing state of the art ([26,27]). The model’s capacitors simulate the cell membrane’s behavior, whereas the resistors represent the current flow in both extracellular and intracellular spaces. The circuit was designed to optimally fit the impedance data while concurrently minimizing its complexity, defined by the number of components, and simplifying the biological interpretation of the impedance output.

### 3.3. Practical Significance of the Results

Differences in impedance and phase angle between fruits and vegetables may result from their diverse tissue structures and water content. Apples had the highest impedance values, which can be explained by their thick skin and relative water content compared to other tested products. In the case of bananas, low impedance values could be due to their soft structure and higher membrane permeability. Pumpkin and potato showed moderate changes, highlighting the need to consider the individual characteristics of each type of product when interpreting the result.

The study results confirm that BIS can be used as a non-destructive and precise method for assessing the freshness of fruits and vegetables. Low frequencies (<1 kHz) proved particularly significant in detecting freshness changes. Additionally, the introduction of advanced phase angle analyses expands the applicability of this method to products that are difficult to assess visually, such as potatoes or pumpkins.

## 4. Discussion

Fruit freshness has become an important research issue, which can be seen in the rising number of publications regarding the freshness of apples [12,28,29], oranges [30], kiwis [31], nectarines [32], strawberries [33,34], bananas [13,35], mangos [36], and others [37,38].

In our study, similar trends to those described by Ibba et al. [13] and Islam et al. [37] were observed; there was a decrease in impedance as the ripening of selected fruits and vegetables progressed. The statistically significant decrease in impedance for pumpkin (*p* = 0.0005) and potato (*p* = 0.0007) determined during the ripening of the products confirms the observed trend. Thus, the BIS method can detect the gradual deterioration of food products by monitoring the decrease in impedance, which correlates with physicochemical changes such as water loss and cell membrane damage. Moreover, a statistically significant increase in phase angle was observed for both pumpkin (*p* = 0.03) and potato (*p* = 0.045), suggesting that cell membranes have become more permeable as fruits and vegetables mature. This supports the contention that BIS allows for monitoring cell membrane changes, a critical food freshness indicator. Increased phase angle values indicate structural changes in the internal tissue, possibly related to ripening.

One of the reasons why BIS has not yet been widely implemented for fruit quality evaluation might be the necessity of developing custom measurement devices based on microcontrollers, as demonstrated by Ibba et al. [13] and Islam et al. [37], who utilized the same AD5933 evaluation board in their studies. However, the limitation of this unit lies in its relatively narrow frequency range of 1 kHz to 100 kHz. This study identified that low frequencies, specifically those below 1 kHz, are particularly crucial for estimating the ripeness of fruits and vegetables. The Analog Discovery 3 employed in this investigation validated its suitability as an impedance measurement equipment, enabling measurements across an extensive frequency spectrum (20 μHz to 25 MHz), thereby obviating the necessity for custom-developed solutions. Furthermore, the device’s minimal mass (~130 g) and compact dimensions (10 cm × 10 cm) present additional advantages.

A significant challenge associated with the BIS method is the need for precise electrode attachment for measurements to ensure accurate and reliable data. Variability in the contact between the electrode and the skin can lead to inconsistent measurements and affect the validity of the results. This makes the solution impractical, especially in commercial applications. To overcome these challenges, advances in flexible and conformal electrode materials, such as textile electrodes [39] that offer better adaptability to different sample shapes, are needed. The development of automated electrode placement and calibration systems can enhance consistency and reduce the need for specialized training. Implementing geometric correction algorithms and advanced signal processing techniques can also mitigate the effects of electrode placement variability, ensuring more reliable and accurate BIS measurements in commercial applications [40,41]. Therefore, further advancements in non-invasive measurement techniques are essential for the broader adoption of BIS in the food industry.

In this study, the scope of BIS measurements was expanded beyond fruits like apples and bananas to include vegetables such as pumpkin and potato, allowing for a more comprehensive evaluation of the method’s applicability in food quality assessment. Previous studies have similarly demonstrated the efficacy of BIS in monitoring ripeness and freshness in various fruits and vegetables, such as tomatoes, kiwis, and strawberries, showing a consistent relationship between impedance variations and physicochemical changes during ripening [11,12,13,31,33]. Differences in impedance and phase angle values observed across different food types in this study align with findings from earlier research, which attributes such variations to structural and compositional differences, including water content and cellular integrity [10,12,28,35]. Our results reinforce the thesis that BIS is a versatile method capable of adapting to a wide range of food produce, offering precise evaluations of freshness. Despite differences in measurement setups and frequency ranges across studies, BIS has consistently proven effective in assessing food quality [13,14,17]. Although red pepper was excluded from our study due to structural instability, this does not undermine the overall efficacy of the BIS method. However, product-specific modifications, such as optimized electrode placement techniques and frequency adjustments, may be necessary to improve measurement reliability for certain food types.

Despite bioimpedance spectroscopy being a promising technique for assessing the freshness of fruits and vegetables by measuring their electrical properties, it has several limitations, especially when considering optimal and realistic product storage conditions. The ideal storage conditions for most fruits and vegetables are a temperature range of 32 °F to 55 °F (0 °C to 13 °C) and high relative humidity (80% to 95%) [42]. However, grocery stores and supermarkets often do not meet optimal storage conditions. Temperature and humidity fluctuations can alter the electrical properties of the product, making it challenging to obtain consistent results [43]. Higher temperatures can decrease the electrical impedance of biological tissues, including fruits and vegetables, because increased temperature enhances ion mobility, reducing resistance and capacitance. As a result, BIS measurements may show lower impedance values, potentially leading to false assessments of freshness. Lower temperatures can increase the electrical impedance as ion mobility is reduced. This can result in higher impedance readings, which may not accurately reflect the actual freshness of the produce. Insufficient humidity can also impact electrical properties due to the dehydration of fruits and vegetables. Dehydration increases resistance and decreases capacitance, resulting in higher impedance values [43]. Excessive humidity can foster the growth of mold and bacteria, which can alter the product’s electrical properties. Maintaining optimal temperature and humidity conditions is crucial for accurate BIS assessments of the freshness of fruits and vegetables. Deviations from these conditions can lead to variations in the electrical properties of the product, resulting in unreliable measurements.

The use of AI and ML for BIS data analysis has the potential to enhance the precision of product classification and quality prediction, making it a valuable tool for automating quality control in the food supply chain. Previous studies have demonstrated that ML algorithms can effectively interpret impedance data to assess ripeness and freshness in fruits such as apples and bananas [12,33]. Similarly, AI-driven models have been applied to optimize storage conditions by predicting changes in water content and enzymatic activity based on bioimpedance measurements [37]. Expanding the frequency range beyond the standard BIS scope could further refine these correlations, improving predictive accuracy for different produce types. In our study, the observed impedance and phase angle variations across different fruits and vegetables suggest that integrating AI-based analysis could enhance the interpretation of these results, especially for products like potatoes and pumpkins, where external ripeness indicators are less apparent. Additionally, implementing BIS in automated production lines could standardize food quality assessments, reducing costs and ensuring consistency at every stage of processing and distribution [13]. Given the growing interdisciplinary interest in BIS, its applications extend beyond food technology into medical and bioengineering fields, reinforcing its relevance in public health and nutrition research.

## 5. Conclusions

The results of this study provide strong statistical evidence supporting the use of bioimpedance spectroscopy for assessing the freshness of various fruits and vegetables. Key parameters such as impedance and phase angle proved to be reliable indicators of physicochemical changes during the ripening process, with statistically significant results observed for pumpkin and potato. The BIS method demonstrated particular effectiveness in identifying subtle structural changes, even in thick-skinned vegetables, which are often challenging to assess visually. Despite the exclusion of red pepper due to its structural instability, this limitation highlights the broader challenge of electrode placement and the need for robust measurement protocols. Future research should focus on developing more flexible electrode systems and automated calibration techniques to improve data consistency. Additionally, expanding the study to include a wider range of food types with varying textures and compositions would enhance the generalizability of the findings. A key area for improvement is the integration of artificial intelligence and machine learning algorithms to facilitate automated data analysis and improve the predictive accuracy of BIS measurements. Such advancements could enable real-time quality assessments in automated production lines, offering a cost-effective and scalable solution for food quality control. Further investigations into the impact of frequency ranges on different food types and the development of AI-based classification models would significantly contribute to the broader adoption of BIS in the food industry. By addressing these challenges and leveraging technological innovations, BIS could become a valuable tool not only for food quality assessment but also for interdisciplinary applications in medicine, bioengineering, and public health.

## Figures and Tables

**Figure 1 foods-14-00947-f001:**
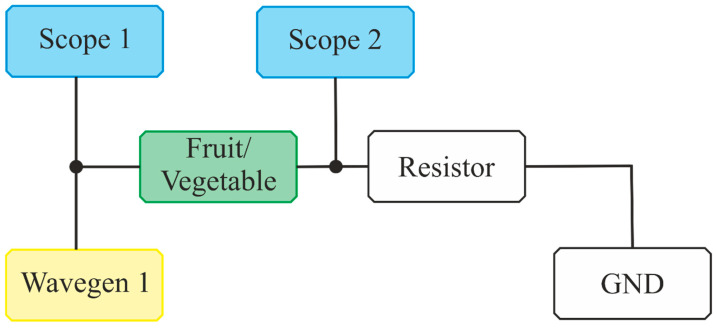
The hardware setup scheme for impedance measurements.

**Figure 2 foods-14-00947-f002:**
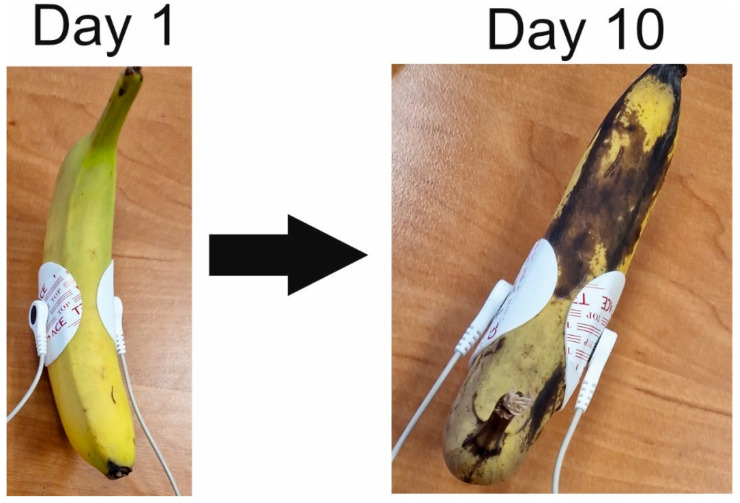
Changes in the appearance of the banana during impedance measurements.

**Figure 3 foods-14-00947-f003:**
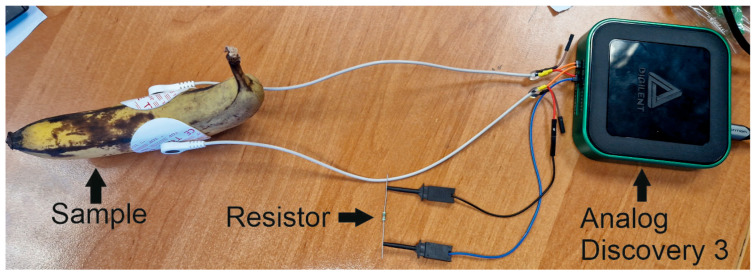
Experimental setup of impedance measurements.

**Figure 4 foods-14-00947-f004:**
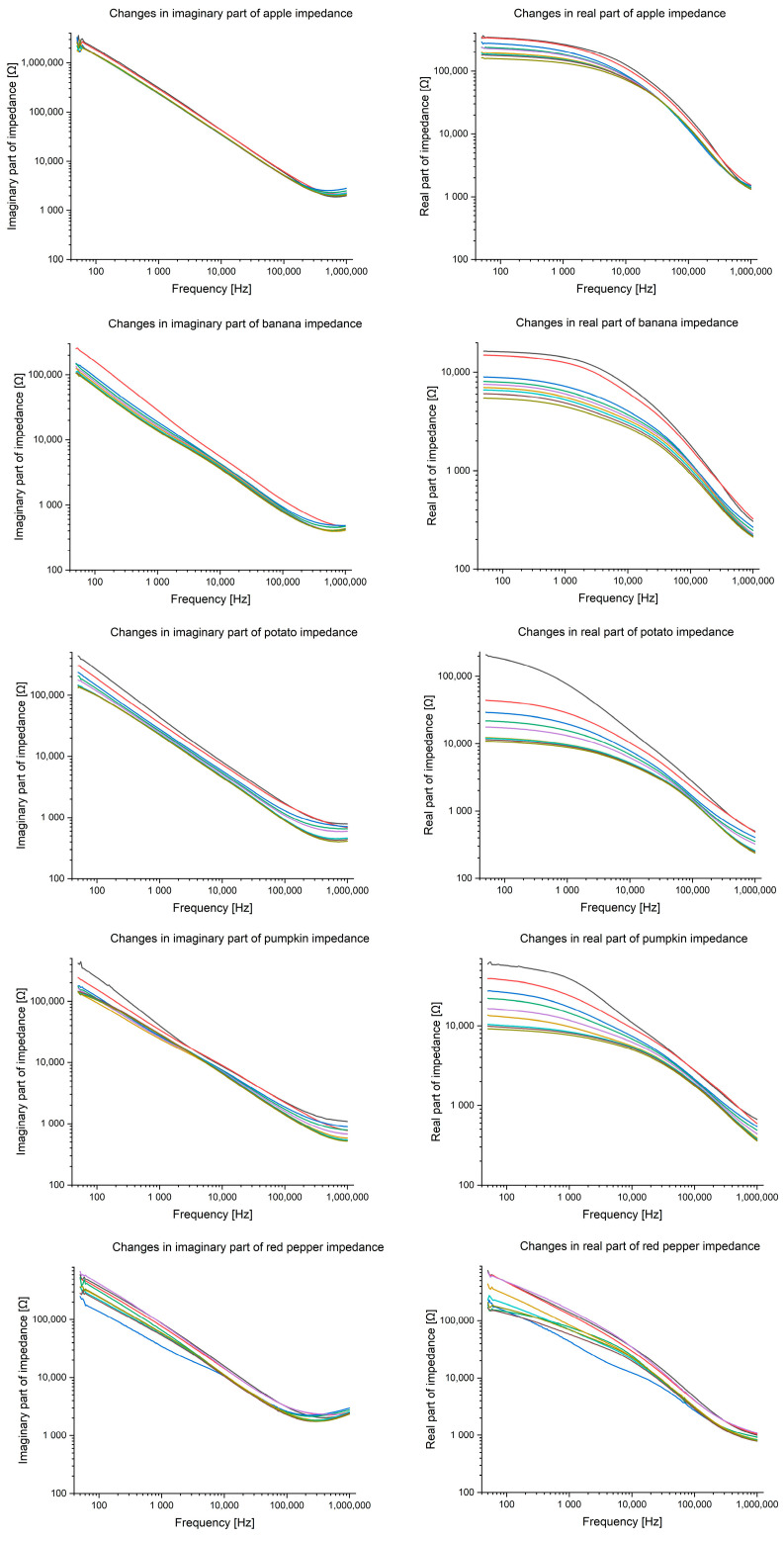
Impedance changes on the days of research for each fruit and vegetable (black—20 November 2023; red—21 November 2023; blue—22 November 2023; green—23 November 2023; purple—24 November 2023; dark yellow—27 November 2023; turquoise—28 November 2023; dark purple—29 November 2023; rotten green—30 November 2023).

**Figure 5 foods-14-00947-f005:**
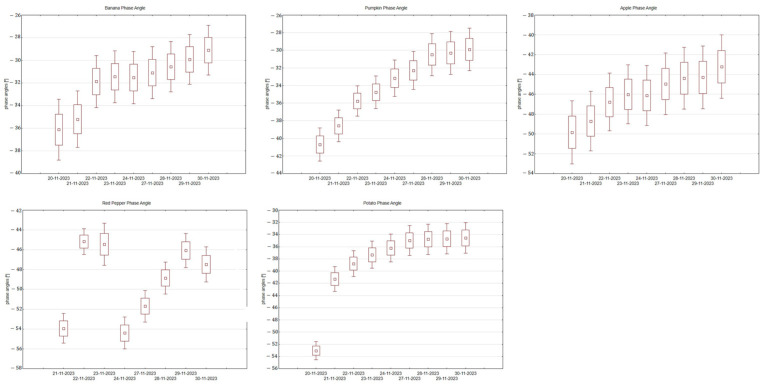
Phase angle changes during the study days for each fruit and vegetable.

**Figure 6 foods-14-00947-f006:**
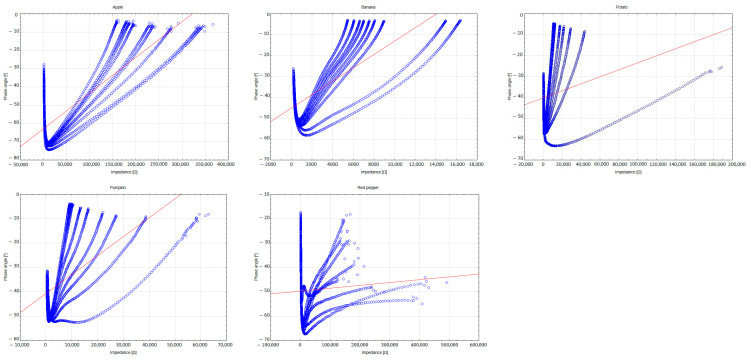
Dependence of phase angle on impedance for various fruits and vegetables (red line—linear fit).

**Figure 7 foods-14-00947-f007:**
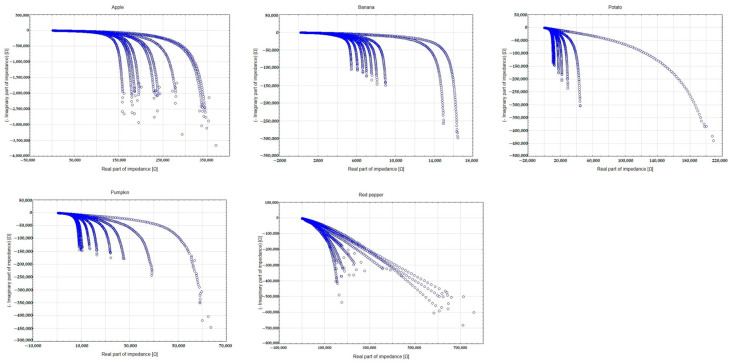
Bode plots for all the vegetables and fruits tested.

**Figure 8 foods-14-00947-f008:**
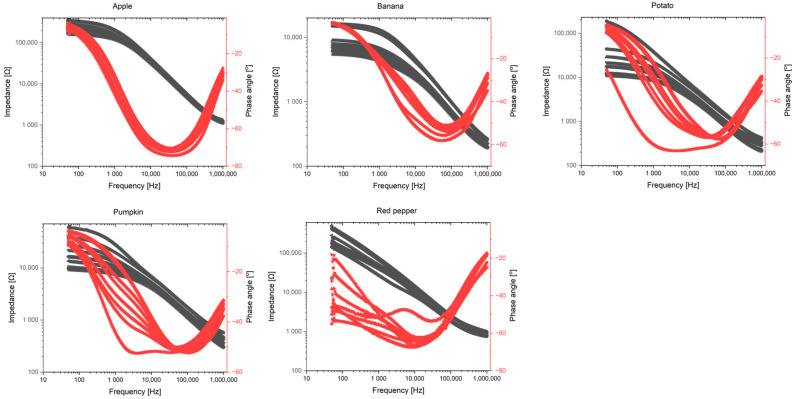
Nyquist plots for all vegetables and fruits tested (black—Impedance; red—Phase angle).

**Figure 9 foods-14-00947-f009:**
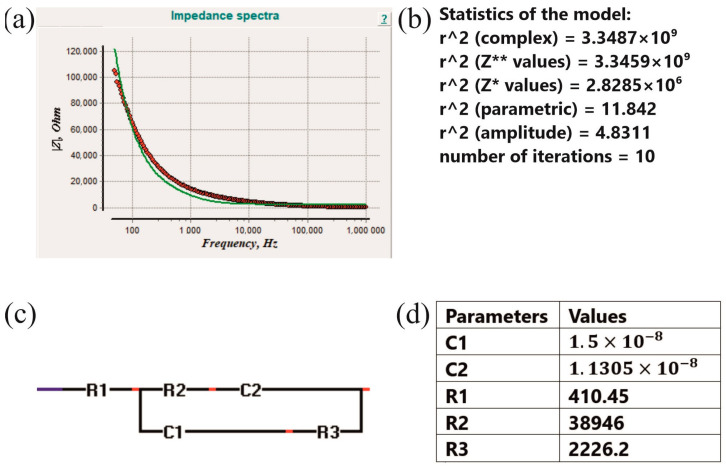
The plot of the impedance data (red line) with the best fit model (green line) (**a**), the statistics of the model (**b**), the equivalent electrical circuit (**c**), and the parameters and their values (**d**). Z* ICG first derivative, Z** ICG second derivative.

**Table 1 foods-14-00947-t001:** The impedance values of banana, pumpkin, apple, potato, and red pepper for selected frequencies: 100 Hz, 1 kHz, 10 kHz, 100 kHz, and 1 MHz.

	Date of Measure-ments	20 November 2023	21 November 2023	22 November 2023	23 November 2023	24 November 2023	27 November 2023	28 November 2023	29 November 2023	30 November 2023	Frequency
Sample											
Banana		16,130	14,677	8785	7910	7408	6868	6490	5949	5357	100 Hz
		12,979	11,392	6708	6023	5635	5227	4925	4578	4228	1 kHz
		4569	4144	2967	2801	2671	2542	2414	2284	2177	10 kHz
		968	955	728	701	679	656	636	607	588	100 kHz
		260	265	233	218	201	194	196	193	190	1 MHz
Pumpkin		55,955	36,353	25,619	20,886	15,699	12,700	9988	9526	8910	100 Hz
		27,568	19,452	14,396	12,537	10,607	8993	8032	7794	7371	1 kHz
		6810	6494	5280	5002	4701	4305	4251	4141	3990	10 kHz
		1727	1704	1390	1304	1231	1145	1134	1109	1077	100 kHz
		568	469	462	412	367	323	309	298	294	1 MHz
Apple		1,951,974	1,833,875	1,467,801	1,444,473	1,434,312	1,437,876	1,453,127	1,483,855	1,456,841	100 Hz
		307,508	288,505	231,104	231,055	228,579	230,109	233,504	237,209	239,628	1 kHz
		43,109	42,797	34710	34,998	34,363	34,505	34,730	35,040	35,698	10 kHz
		5778	6044	5133	5113	4978	4969	4994	5009	5097	100 kHz
		2000	2247	2825	2511	2306	2190	2229	2091	2085	1 MHz
Potato		257,728	186,485	142,303	126,359	116,337	98,624	100,205	99,473	98,201	100 Hz
		42,915	34,604	27,372	25518	24,346	22,565	22,767	22,380	22,047	1 kHz
		8145	7377	5802	5390	5115	4708	4717	4606	4515	10 kHz
		1689	1637	1280	1138	1069	911	909	890	871	100 kHz
		783	683	704	651	596	461	462	439	412	1 MHz
Red Pepper		376,420	342,961	137,915	309,635	414,642	242,863	220,482	208,048	247,291	100 Hz
		84,965	75,313	34,125	64,219	83,917	57,117	54,747	52,892	57,955	1 kHz
		15,814	14,329	10,560	10,903	14,629	11,199	10,904	10,545	11,061	10 kHz
		3065	3126	2563	2518	3123	2407	2366	2265	2336	100 kHz
		2417	2550	2977	2762	2338	2534	2570	2455	2343	1 MHz

## Data Availability

The raw data supporting the conclusions of this article will be made available by the authors on request.

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
