# Peer review of "An Assessment of the Freshness of Fruits and Vegetables Through the Utilization of Bioimpedance Spectroscopy (BIS)—A Preliminary Study"

_foods, 2025, doi:10.3390/foods14060947_

Round 1
Reviewer 1 Report
Comments and Suggestions for Authors
Title: Enhancing the Assessment of Freshness in Fruits and Vegetables through the Utilization of bioimpedance Spectroscopy (BIS)
Comments: Authors reported on the use of Bioimpedance spectroscopy as a non-invasive method to assess the freshness of the fruits and vegetables. Though the work is interesting in evaluating the food quality but the results are not convincing to adopt this proposed method as a standalone technique for food quality assessment. There are many drawbacks in this proposed technique and authors need to provide sufficient results to convince this reviewer that this proposed method is used for Food quality measurements. Therefore, I recommend a MAJOR revision to this manuscript and consider the following specific comments before resubmitting their revised version.
Specific comments:
- A digital picture of the whole experimental set-up should be included.
- Authors should have repeated varieties of same fruit and vegetables starting from unripe state to rotten state.
- Table 1 should be modified; “date of measurement” row should be included and also error bars or error % for all the values must be included.
- Figure 4: Extract the real and imaginary parts of impedance and normalize the data to show variations in Z. The current figure 4 is hard to evaluate the changes in Z in all measurements.
- Phase angle w.r.t impedance plots should be added and also the impedance data should be fit with equivalent circuit and extract the parameters.
- There is still lacks some results to convince this reviewer as the proposed method to best suited for food quality, authors are encouraged to add more characterization results to the revised manuscript.
-
Reviewer 2 Report
Comments and Suggestions for Authors
1. Keywords: Fruit is changed to: Fruits; add a keyword: Freshness
2. Line 30-32: There is an error in the expression of the sentence and it needs to be revised.
3. References cited in the text should be numbered in the order of appearance according to convention. Please refer to the requirements of the journal and complete the modification.
4. Line 78, “1.1 Related works” This section is inappropriate to appear here alone. It is recommended to merge it with the previous introduction section according to the correct logical order to facilitate readers' reading and understanding.
5. 2. Materials and Methods 2.1. Preparations of materials This section needs to supplement the specific information such as the specific variety, harvesting time, harvesting location, etc. of the experimental materials (pumpkin, red pepper, potato, banana and apple).
6. line 136: It is recommended to change the title of section 2.4 to “Statistical analysis”.
7. The image in figure 4 is too blurry. The author should modify it to a clearer data graph.
8. Figure 5 is also too blurry. The author must improve the clarity of the data graph. In addition, the author confuses the word "figure" with "Fig" in the figure title and the text. Please check the entire text and make the changes.
9. The article is novel and innovative, but some very important information is missing in the experimental design, such as how many samples were tested for each sample? Because this is the freshness of the sample being tested, the number of samples must be large enough and representative. In addition, the article does not discuss the research results in depth, and the highlights of this study are not shown compared to the current research. In summary, I think the author can be given a chance to revise the article.
Comments on the Quality of English LanguageThe quality of english language should be improved.
Reviewer 3 Report
Comments and Suggestions for Authors
This study investigates bioimpedance spectroscopy (BIS) as a non-invasive method for assessing fruit and vegetable freshness by measuring impedance and phase angle changes over nine days in potatoes, pumpkins, red peppers, apples, and bananas using the Analog Discovery 3 device across frequencies from 50 Hz to 1 MHz. The study results indicate a consistent decrease in impedance and an increase in phase angle during ripening, with statistically significant changes in pumpkin and potato. The study revealed that despite structural challenges in red pepper, BIS proves to be an effective, objective, and non-destructive alternative to traditional chemical methods.
General comments and queries on the study (Some might be included in the attached PDF):
· What was the rationale behind selecting the specific fruits and vegetables (potato, pumpkin, red pepper, apple, and banana)?
· The selected fruits and vegetables were stored at 16°C and 35% humidity. Is there any reference supporting these as optimal storage conditions for all the tested commodities?
· Would different storage conditions affect impedance measurements?
· Since the produce was purchased from a local store, how was the freshness at the time of purchase ensured? I guess that results might differ if samples were collected directly from the farm?
· Red pepper was removed from the study due to structural damage. Could this challenge be mitigated through an alternative method or electrode placement? Could this be due to its high-water content or different cellular structure?
· Why was the 50 Hz to 1 MHz frequency range selected?
· Since electrode placement can influence impedance measurements, how was this controlled to ensure consistency across different samples and shapes?
· The study mentions a significant increase in phase angle correlating with ripening. However, could external factors such as temperature fluctuations or dehydration contribute to this change?
I have suggested some changes in the attached PDF, which are self-explanatory.

The English language needs improvement.
Round 2
Reviewer 1 Report
Comments and Suggestions for Authors
The authors have addressed all of the comments raised by this reviewer, and I have no further comments to make on this revised manuscript, and thus it is now acceptable for publication in this journal in its current format.
Author Response
We have modified, and added additional text from lines 320-340 which is a response to Academic Editor.
The authors' response to the reviewer is misleading because in fact none of the papers they refer to present such storage conditions. It is plausible that these conditions were chosen for better performance of their equipment in order to prevent the possibility of water condensation that would disturb the analysis, but this just questions the feasibility of their approach. The problems probably might be alleviated if the produce would be kept in realistic storage environment and transferred to low humidity conditions just for performing the analysis, with providing time necessary for temperature and humidity equilibration. Anyhow, the storage of potatoes, apples etc. lasts much more than 9 days, and this scheme is inevitable if the authors seriously consider the application of their instrument. In spite of these problems, I respect the acceptance recommendations of the reviewers. However, serious and sincere addressing of the above limitations in the discussion is essential; otherwise, the manuscript would mislead the readers.
Answer:
It is true that we do not discuss in detail the limitations of using the bioimpedance spectroscopy in assessing the freshness of fruits and vegetables. We have added a paragraph in the article that describes the limitations of the BIS method in context to the changes in temperature and humidity that occur during storage and distribution of fruits and vegetables. We would also like to emphasize that the food storage conditions we maintained were chosen only for the experiment. Maintaining stable temperature and humidity conditions is important for accurate BIS measurements. We also supplemented the text by adding relevant references.
Answer:
Another point, I disagree with the revised title of the manuscript: "Enhancing the Assessment" and "Enhancing the Freshness" are two completely different things. The trials presented have nothing to do with "enhancing freshness"; opposite, they enhance the deterioration. I propose a title without any "enhancement": "Assessment of Freshness of Fruits and Vegetables through the Utilization of Bioimpedance Spectroscopy (BIS) - a Preliminary Study".
The title has been changed as proposed.
Reviewer 2 Report
Comments and Suggestions for Authors
Accept in present form
Author Response
We have modified, and added additional text from lines 320-340 which is a response to Academic Editor.
The authors' response to the reviewer is misleading because in fact none of the papers they refer to present such storage conditions. It is plausible that these conditions were chosen for better performance of their equipment in order to prevent the possibility of water condensation that would disturb the analysis, but this just questions the feasibility of their approach. The problems probably might be alleviated if the produce would be kept in realistic storage environment and transferred to low humidity conditions just for performing the analysis, with providing time necessary for temperature and humidity equilibration. Anyhow, the storage of potatoes, apples etc. lasts much more than 9 days, and this scheme is inevitable if the authors seriously consider the application of their instrument. In spite of these problems, I respect the acceptance recommendations of the reviewers. However, serious and sincere addressing of the above limitations in the discussion is essential; otherwise, the manuscript would mislead the readers.
Answer:
It is true that we do not discuss in detail the limitations of using the bioimpedance spectroscopy in assessing the freshness of fruits and vegetables. We have added a paragraph in the article that describes the limitations of the BIS method in context to the changes in temperature and humidity that occur during storage and distribution of fruits and vegetables. We would also like to emphasize that the food storage conditions we maintained were chosen only for the experiment. Maintaining stable temperature and humidity conditions is important for accurate BIS measurements. We also supplemented the text by adding relevant references.
Another point, I disagree with the revised title of the manuscript: "Enhancing the Assessment" and "Enhancing the Freshness" are two completely different things. The trials presented have nothing to do with "enhancing freshness"; opposite, they enhance the deterioration. I propose a title without any "enhancement": "Assessment of Freshness of Fruits and Vegetables through the Utilization of Bioimpedance Spectroscopy (BIS) - a Preliminary Study".
Answer:
The title has been changed as proposed.
Reviewer 3 Report
Comments and Suggestions for Authors
I appreciate the effort the authors have put into revising their manuscript and addressing the concerns raised in the review process. Upon reviewing the revised version, I find that authors have satisfactorily responded to all the queries and incorporated the necessary improvements.
The revisions have significantly enhanced the clarity and quality of the paper, and I am now satisfied with its current form. I have no further comments or suggestions, and I believe the manuscript is ready for acceptance.
Author Response

(The authors gave the same response as above.)
